# Estimating conformational landscapes from Cryo-EM particles by 3D Zernike polynomials

D. Herreros [1] ✉, R. R. Lederman [2], J. M. Krieger[1], A. Jiménez-Moreno[1], M. Martínez [1], D. Myška[3], D. Strelak[1,4], J. Filipovic [3], C. O. S. Sorzano [1,5] & J. M. Carazo [1,5]

The new developments in Cryo-EM Single Particle Analysis are helping us to understand how the macromolecular structure and function meet to drive biological processes. By capturing many states at the particle level, it is possible to address how macromolecules explore different conformations, information that is classically extracted through 3D classification. However, the limitations of classical approaches prevent us from fully understanding the complete conformational landscape due to the reduced number of discrete states accurately reconstructed. To characterize the whole structural spectrum of a macromolecule, we propose an extension of our Zernike3D approach, able to extract per-image continuous flexibility information directly from a particle dataset. Also, our method can be seamlessly applied to images, maps or atomic models, opening integrative possibilities. Furthermore, we introduce the ZART reconstruction algorithm, which considers the Zernike3D deformation fields to revert particle conformational changes during the reconstruction process, thus minimizing the blurring induced by molecular motions.

Cryo-electron microscopy (Cryo-EM) single particle analysis (SPA)[1] has proven to be a powerful technique to understand the structure of macromolecules. By capturing individual images of the specimen in different poses, it is not only possible to reconstruct the average macromolecular conformation of the specimen under study, but it also brings to light the challenging problem of identifying several conformational states from the acquired dataset.

Generally, compositional heterogeneity, as well as flexibility, have been addressed through 3D classification[2]. This approach allows reconstructing a given number of different states from the particle images based on the assumption that there is a defined number of discrete conformational states being explored by the specimen. This methodology has been very successful in the study of many systems, being recently expanded to increase the number of states being resolved[3].

However, the explicit modeling assumption of the existence of discrete motions has obvious limitations in most experimental cases, depending on the actual biological system under study. Clearly, removing this constraint is methodologically very challenging, although the pay-offs are clear, both in terms of obtaining richer conformational landscapes than currently done, and in providing improved algorithmic stability and objectivity, removing many assumptions and trial and error tests.

Limitations faced with discrete flexibility can only be solved at the image processing level by a paradigm change introducing methods able to handle continuous flexibility: the ability to extract macromolecular conformational information at the individual particle level to get a sufficiently rich and populated landscape of molecular states. Several approaches have been previously proposed to face continuous flexibility, each from a different perspective[4–9].

[1]Centro Nacional de Biotecnologia-CSIC, C/Darwin, 3, 28049 Cantoblanco, Madrid, Spain. [2]The Department of Statistics and Data Science, Yale University, New Haven, CT, USA. [3]Institute of Computer Science, Masaryk University, Botanická 68a, 60200 Brno, Czech Republic. [4]Faculty of Informatics, Masaryk University, Botanická 68a, 60200 Brno, Czech Republic. [5]These authors jointly supervised this work: C.O.S. Sorzano, and J.M. Carazo. ✉e-mail: dherreros@cnb.csic.es

In this work, we extend our recent Zernike3D algorithm[10] (specifically designed to deal with continuous heterogeneity) to precisely accomplish the latter task starting from Cryo-EM images with some unique properties, such as (1) the possibility to work with images, maps, and atomic models in the same space, (2) a clear mathematical design that intrinsically helps avoiding over-deformations in projection directions, and (3) a reconstruction algorithm (that we name ZART - Zernike3D-based Algebraic Reconstruction Technique) that takes into account individual particle conformational information, reverts the structural changes, and obtains a new map in which flexible regions have intrinsically increased resolution. Note that property (1) indicates that one can work at the level of structural models, avoiding multiple fitting steps, property (2) drastically reduces flexibility estimation errors that would be very difficult to consider in other mathematical frameworks, and property (3) makes it possible to explore states with a small number of classes while still reconstructing maps with large datasets, though at improved resolution since motion blurring is substantially reduced.

We note that the full derivation of ZART is rather technical, so we present in this work its main properties in the context of continuous flexibility, while the derivation of the algorithm in itself is presented as a separate technical work.

## Results

### Conformational landscape of EMPIAR-10028 dataset

The following experiment is aimed at assessing the capacity of the Zernike3D algorithm to identify conformational variability on real Cryo-EM data. To that end, we analyzed the EMPIAR-10028 dataset[11] corresponding to the *P. falciparum* 80 S ribosome bound to emetine. This dataset has been extensively studied by other methods[5,6], becoming a popular validation dataset for continuous heterogeneity algorithms.

In this work we have reprocessed that dataset inside Scipion[12], leading to a total of 50,000 particles. The workflow followed included several cleaning steps to reduce as much as possible the number of unwanted particles, followed by some consensus protocols to compare the parameters estimated by different algorithms (angular assignation, shifts, Contrast Transfer Function…) and keep only the particles consistently estimated.

The previous particles were subjected to the Zernike3D analysis, translating them to a set of Zernike3D coefficients. The maximum basis degrees were set to $N = 3$ and $L = 2$ for the estimation of the deformation fields. In addition, the particles were downsampled to a box size of 125 voxels to increase the performance of the algorithm. Apart from the Zernike3D analysis, the particles were not subjected to other heterogeneity workflows such as classical 3D classification.

The resulting UMAP (Uniform Manifold Approximation and Projection)[13] representation of the Zernike3D coefficient space is shown in Fig. 1. As it can be seen from the representation, the Zernike3D coefficient space leads to an informative representation of the heterogeneity present in the dataset. Two clear states are well differentiated, representing the two rotation states of the small subunit of the ribosome, as well as some other more localized movements. The colormap used to represent the embedding describes the amount of deformation associated with each deformation field: purple colors correspond to small deformations and are usually associated with conformational changes similar to the reference map, while yellow colors are associated with bigger changes. The possibility of coloring the coefficient space adds another dimension of information helping in the analysis of the heterogeneity of the dataset.

There are two different possibilities to recover conformational changes from the previous embedding: (1) applying a deformation field to the reference or (2) exploring, by refinement and reconstruction, different areas of the conformational space. Option 1 represents an almost instant and interactive exploration of the conformational

space, in which just by placing the cursor on any point of the representation conformation we obtain a Zernike3D synthesis of a map, while Option 2 goes to the original images and aims at exploring whether there are residual errors not accounted for by Zernike3D. In all cases tested so far, the differences between the two options are minimal, as it is shown in Fig. 2a, b. However, the application of the deformation field leads to a higher resolution representation of the conformational change (equal to the resolution of the reference map), while the refinement resolution is intrinsically limited to the number of particles selected from the space.

In addition, the Zernike3D coefficients extracted from the conformational space can be applied simultaneously to the reference map and to a structural model traced (or aligned) from the reference. This allows obtaining a rigid fitting of the atomic positions that match the conformation of any particle in the dataset. An example of the application of the Zernike3D coefficients to the ribosome atomic structure can be found in Fig. 2c. However, it is worth mentioning that the Zernike3D coefficients are computed exclusively based on geometrical considerations, so the approximated structural models might need to be refined to correct for stereochemistry artifacts. Indeed, it should always be considered that the estimation of the deformation fields describing a given transition only depends on the rigid alignment of the reference towards the conformation represented by a given particle. Therefore, the estimated deformation field does not consider any stereochemistry constrains, which should be posteriorly imposed to avoid atomic clashes or improve Ramachandran outliers among others.

An example of the simultaneous exploration of the coefficient space performed with the reference map and its structural model is provided in Supplementary Movie 1 (we are aware that this and subsequent videos are only graphical means to make more obvious conformational changes, and that they are not to be considered as suggesting molecular trajectories at all). The different states were obtained by grouping the coefficients with KMeans into 5 clusters. Then, the cluster representatives were used to generate the deformed maps/structures, which were afterward morphed with ChimeraX software[14].

The next step we followed in the analysis of the dataset is to use the estimated deformation fields and the particles to reconstruct a higher-resolution map by correcting the conformational changes of each image with ZART. The comparison between the map reconstructed with CryoSparc[15] and ZART reconstruction algorithm is shown

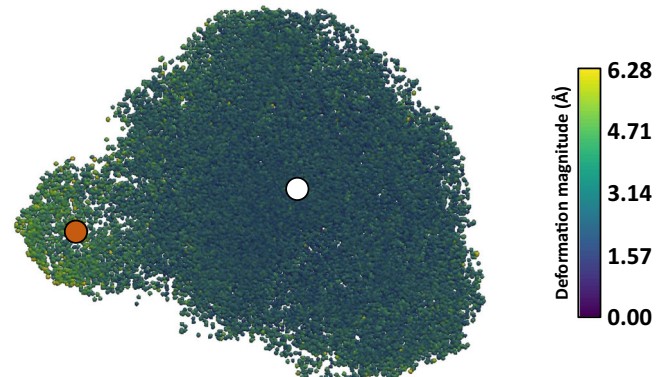

**Fig. 1 | EMPIAR-10028 Zernike3D conformational landscape.** UMAP representation of the Zernike3D coefficient space for the P. falciparum 80 S ribosome (EMPIAR-10028 dataset). The colormap represents the modulus of the deformation field that has to be applied to the reference map to match the conformational state of each particle projection image. Purple colors represent lower deformations (close to the reference state). The representation shows a clear distinction between two different states marked by the white (reference map) and orange (rotated Pf80S state) dots.

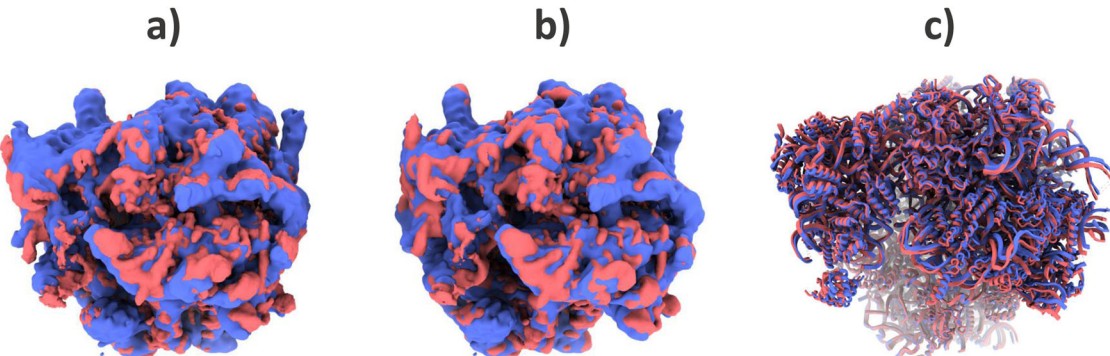

**Fig. 2 | Example of Pf80S Zernike3D states. a** Comparison of the reference conformation required by the Zernike3D algorithm (red) and the rotated Pf80S state recovered from a homogeneous refinement with CryoSparc (blue). **b** Comparison of the rotated Pf80S state recovered from the Zernike3D deformation fields (red) and the rotated Pf80S state recovered from a homogeneous refinement with CryoSparc (blue). The particles processed by CryoSparc are taken from the coefficient space area defined by the orange dot in Fig. 1, and the deformation field is computed with the coefficients associated with this dot. The comparison between the maps displayed in **a** and **b** show that the Zernike3D conformation (**b** - red map) is consistent with the experimental conformation refined from the particles selected from that region of the coefficient space (blue). In addition, the application of the deformation field does not decrease the resolution of the reference map. **c** Comparison of the atomic structure associated with the Zernike3D reference map (red) and the structure deformed with the Zernike3D deformation fields (blue). Since the Zernike3D can work indistinguishably with maps, atomic structures, and particles, the rotated state can be appropriately reproduced at the atomic level using the deformation fields estimated from the particles.

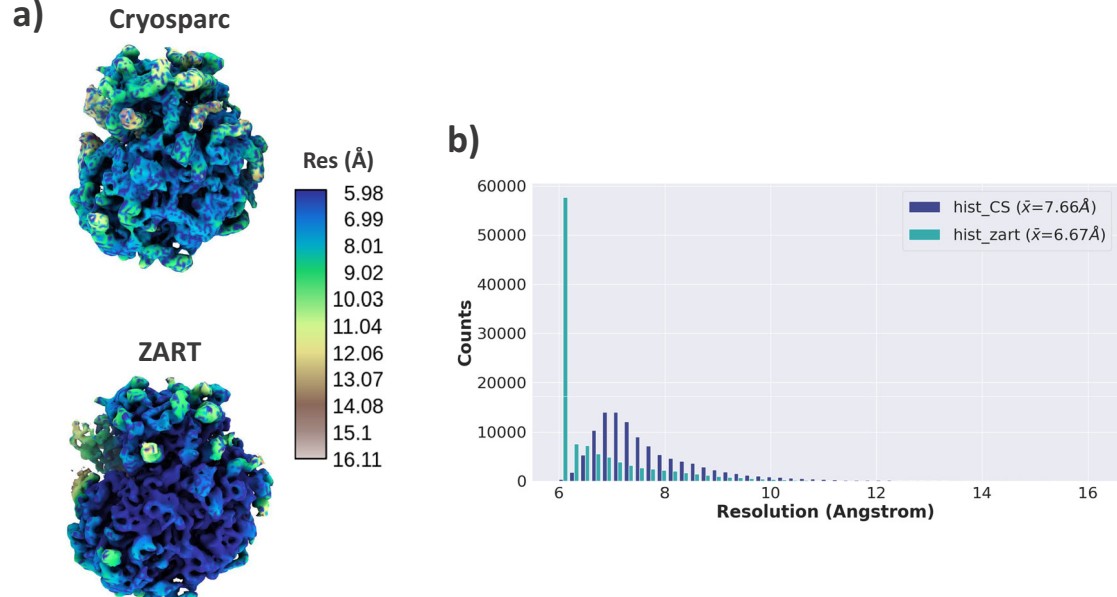

**Fig. 3 | Analysis of EMPIAR-10028 ZART reconstruction. a** Comparison of *P. falciparum* 80 S ribosome map refined with CryoSparc (blue) and the motion-corrected map recover with our ZART algorithm. The colormap represents the local resolution estimation for each voxel computed with BlocRes[23]. The ZART reconstructed map shows an overall improvement in resolution thanks to the deformation fields considered during the reconstruction process. **b** Resolution histogram comparison for CryoSparc and ZART reconstructions obtained from the resolution map computed by BlocRes. The histogram shows a clear displacement of the local resolutions towards higher resolutions. The value provided in the legend of the histograms shows the mean value of the local resolution estimations for both reconstructions.

in Fig. 3a. The comparison of the two maps shows a clear improvement at both, the level of maps (a) and slices (b), in the moving and still areas of the molecule. In order to make a more quantitative comparison of the maps, we computed the local resolution histograms of both reconstructions, which are compared in Fig. 3b. Similarly to the visual inspection of the maps, the resolution histograms confirm the improvement in local resolution, being the average resolution of ZART pushed 1.01 Å compared to the mean resolution of CryoSparc.

### Conformational landscape of EMPIAR-10180 dataset

The EMPIAR-10180[16] dataset has become another standard dataset to test continuous heterogeneity algorithms due to the large degree of flexibility information it contains. The dataset corresponds to a pre-catalytic spliceosome exhibiting an extensive heterogeneity already observed by classical methods such as 3D classification.

Since the Zernike3D algorithm focuses on the analysis of continuous heterogeneity rather than compositional heterogeneity, the dataset was preprocessed inside Scipion[12] to clean as much as possible the original deposited particles. The original dataset is composed of around 320k particles, which were reduced to around 180k after the cleaning steps.

The cleaned particles were afterward subjected to the Zernike3D analysis to extract the different conformational changes suffered by the pre-catalytic spliceosome. As we did in the previous

a)

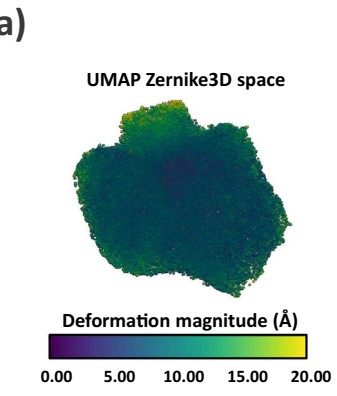

UMAP Zernike3D space

Deformation magnitude (Å)

0.00   5.00   10.00   15.00   20.00

b)

c)

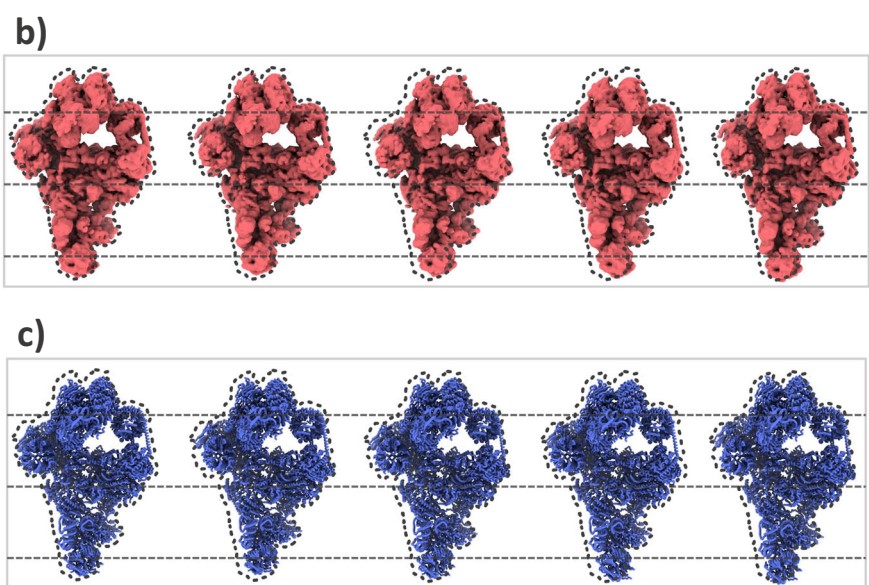

**Fig. 4 | Analysis of the EMPIAR-10180 Zernike3D conformational landscape.** **a** UMAP representation of the Zernike3D coefficient space obtained from the estimation of the per-particle conformational changes associated with EMPIAR-10180. The color map represents the deformation field magnitude associated with each particle involved in the analysis. **b** Example of five conformations extracted from the landscape shown in **a** after clustering by KMeans. The different conformations correspond to the representatives of each KMean cluster. The dotted gray lines are provided to enhance the visualization of the conformational changes. The dotted map corresponds to the reference conformation, provided also to aid in the visualization of the conformational changes. **c** Example of the conformational changes shown in **b** at atomic levels. The conformations were obtained after applying the deformation fields associated with the KMean representatives in **b** to the atomic structure deposited with the EMPIAR-10180 dataset. The dotted lines and the contour of the reference map are also shown to aid in the visualization of the motions.

experiment, we set the maximum basis degrees to $N = 3$ and $L = 2$, and particles were binned to a box size of 128 pixels. The resulting Zernike3D coefficient space is represented in Fig. 4a. The Zernike3D space obtained is similar to the continuous heterogeneity region described by other software like CryoDrgn (Fig. 6 of their manuscript). However, the representation of the conformational changes followed in the Zernike3D approach provides a more versatile manner to assess structural variability.

An example of the versatility of the Zernike3D results is shown in Fig. 4b, c, and Supplementary Movie 2. The maps and structures shown in both Figures were obtained by clustering the Zernike3D space with KMeans into 5 different regions. Then, the representative Zernike3D coefficients of each cluster were extracted to represent the different conformational changes.

Similarly to other algorithms, the conformational changes can be represented at the level of Cryo-EM maps, although the Zernike3D representation will keep the same resolution as the reference map used for the analysis. In addition, the Zernike3D deformation fields can also be applied directly to an atomic structure traced or fitted to the

reference map. In this way, it is also possible to compare the different conformations at an atomic level.

An example of the comparison between two of the previous structures is provided in Fig. 5. Thanks to the Zernike3D approach, it is possible to analyze both, the local and global motion of the atoms present in the structure, which provides a more accurate and informative representation of the conformational changes suffered by the spliceosome.

### SARS-CoV-2 spike one RBD up the conformational landscape
We next applied the Zernike3D algorithm to a set of particles acquired from the SARS-CoV-2 spike. In our previous work[17], we followed a discrete classification approach followed by a PCA (Principal Component Analysis)[18] to study the presence of flexibility in these images, revealing two different open conformations of one of the Receptor Binding Domains (RBDs). The conformations represent small motions around an open RBD state.

The analysis of this dataset is useful to assess the ability of the Zernike3D algorithm to detect small motions from the noisy Cryo-EM

images. Thus, we estimated the deformation fields for each particle starting from one of the conformations reported in ref. [17]. The parameters set for this execution were the same as those used in the previous experiments ($N = 3$ and $L = 2$, yielding a total of 39 components per coefficient set. The particles were also downsampled to a box size of 125 voxels). The UMAP representation of this space is shown in Fig. 6a. The resulting space displays several interesting regions to be analyzed, and it is much richer than the space explored by discrete classification.

In addition, we can integrate the results of the previous discrete classification analysis, resulting in two main classes, with our continuous flexibility approach, by projecting all this information into the same Zernike3D space (in practice, in the reduced representation of the conformational landscape), effectively combining maps and images. The combined space is shown in Fig. 6b. The new representation simplifies the analysis of the embedding, aiding in the identification of the possible conformational changes of the spike by comparing the continuous states to the information of the discrete classification. Clearly, there is much more flexibility than the one originally accounted for by the discrete classification.

An exploration of the conformational space shown in Fig. 6a is provided in Supplementary Movie 3. The different states presented in the video were obtained by applying a set of 20 Zernike3D coefficients

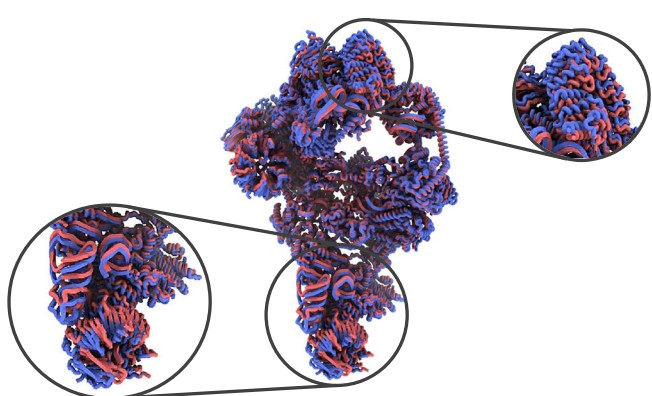

**Fig. 5 | Example of recovered Zernike3D spliceosome states at atomic level.** Example of two conformations obtained from the KMeans clustering of the Zernike3D space in Fig. 4. The conformations were obtained after applying the Zernike3D deformation fields to the atomic structure associated with the reference map used during the Zernike3D analysis. The versatility of the Zernike3D results allows following both, the local and global conformational changes due to the atom's motion, as shown from the different zooms in the recovered structures.

to the reference map and its traced structure, followed by morphing in ChimeraX. The representatives were obtained by clustering the space with KMeans.

The embedding shows an interesting region (composed of a low number of particles) along the direction defined by the white dots representing each classified map. The analysis of this region reveals a conformational change moving in the opposite direction to the one defined by the two discrete classes, which was not previously identified. Supplementary Movie 4 shows the whole motion of the 1Up RBD defined by the main transition identified in the coefficient space. This result shows the importance of analyzing the heterogeneity on a per-particle basis, as discrete classification might not have the ability to resolve low-represented states.

The next step we followed in the analysis of the dataset was to use the estimated deformation fields and the particles to reconstruct a higher-resolution map by "undoing" the conformational changes of each image. The motion-corrected map reconstructed with ZART is provided in Fig. 7a. As expected, the information available in the deformation fields leads to a better resolvability of the moving areas of the spike (the RBDs and N-terminal domains (NTDs) for this specific case), increasing the local resolution of these regions. Fig 7b shows a comparison of the local resolution histograms associated with the maps shown in Fig. 7a. The correction of the per-particle conformational changes leads to a significant increment of the local resolution in the case of ZART, thanks to the reduction of the motion induced blurring present in the CryoSparc reconstruction.

## Discussion

Continuous heterogeneity is widely considered to be a significant breakthrough in the Cryo-EM field, progressively becoming more popular, as shown by the several new software developments to analyze this information from the acquired particle images.

In this regard, we have introduced an extension of the Zernike3D algorithm during this work, which has proven to be a versatile tool to study the continuous motion of macromolecules at the level of maps, structures, and particle images. The extension focuses on the extraction of per-particle conformations, leading to a much more detailed description of the conformational landscape of a molecule compared to classical 3D classification approaches. Furthermore, the versatility of the Zernike3D basis unites maps, particles, and structures into a common framework, opening new possibilities to perform combined heterogeneity analysis with all data available.

Moreover, we have proven that the resulting coefficient space can be applied simultaneously to Cryo-EM maps and atomic coordinates to approximate a new conformational state. The approximation of the

**a)**

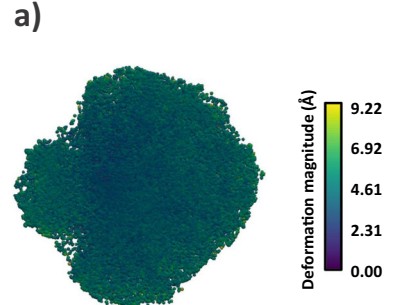

Deformation magnitude (Å)
9.22
6.92
4.61
2.31
0.00

**b)**

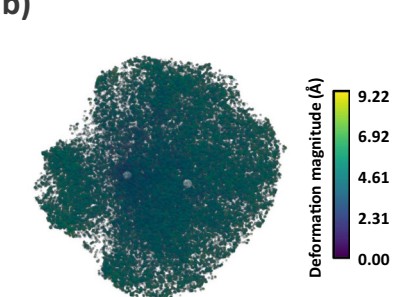

Deformation magnitude (Å)
9.22
6.92
4.61
2.31
0.00

**Fig. 6 | SARS-CoV-2 Zernike3D conformational landscapes. a** UMAP representation of the Zernike3D coefficient space for the SARS-CoV-2 coronavirus spike open state obtained from the particles analyzed previously in ref. [17]. Each point in the space represents a different particle conformation. **b** Combined analysis of particles and volumes (in white dots) corresponding to the two RBD states

described in ref. [17]. Thanks to the combined analysis, we can detect a clear group of particles corresponding to an unidentified conformation in the dataset. The colormap in both images represents the modulus of the deformation field associated with the conformation estimated for every particle.

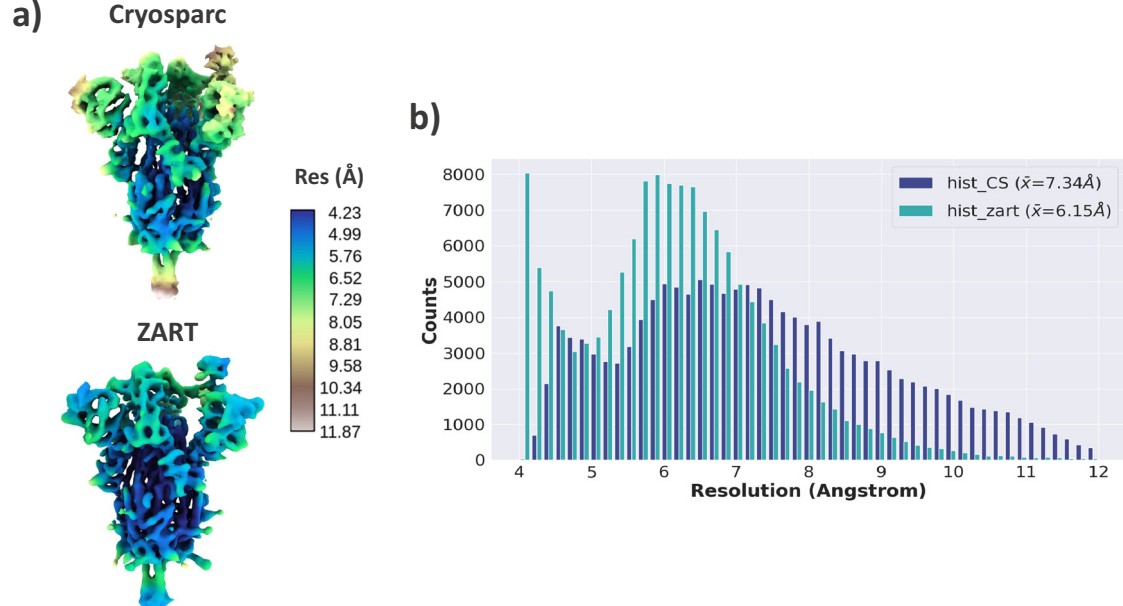

**Fig. 7 | Analysis of SARS-CoV-2 ZART reconstruction. a** Comparison of SARS-CoV-2 coronavirus spike refined with CryoSparc and the motion-corrected map recover with our ZART algorithm. The colormap represents the local resolution computed from BlocRes[23]. ZART reconstruction presents a clear improvement in map quality in the RBD, NTDs, and other regions of the spike thanks to the correction of the motions. **b** Resolution histogram comparison for CryoSparc and ZART reconstructions. The histogram shows a clear displacement of the local resolutions towards the high-resolution regime in the case of ZART. The resolution value provided in the legend of the histograms corresponds to the mean of the local resolution measurements.

conformational changes at the atomic level supposes another step in the connection of the Cryo-EM landscapes with molecular dynamics. This connection will allow getting real energetic landscapes directly based on experimental data in the future.

In addition, we have developed the ZART reconstruction algorithm, which considers deformation fields during the reconstruction to "undo" conformational changes. In this way, it is possible to model the blurring artifacts induced by molecular motions and increase the local resolution of the reconstructed volumes.

## Methods

This section is organized starting with general presentations of the Zernike3D basis and its use for the case of particles exhibiting continuous flexibility (first two subsections), and then dedicating several subsections to useful properties of our proposed method, see also Supplementary Methods.

We also provide some metrics regarding the performance of the Zernike3D algorithm in Table 1.

### Zernike3D basis definition

We use the Zernike3D to estimate the deformation field associated with a given conformational transition, as we previously described in our work[10].

The Zernike3D basis is an infinite-dimensional function space defined over the unit ball. Thus, it is convenient to express it as the combination of a radial and an angular component. For this basis, we have chosen the normalized and generalized definition of the Zernike

polynomials as the radial component:

$$\bar{R}_{l,n}^{p}(x) = \sqrt{2}\sqrt{2n + l + \frac{p}{2} + 1}R_{l,n}^{p}(x) \tag{1}$$

$p$ being a parameter associated with the inner product and dimensionality of the polynomials. For example, in a 3D scenario, the appropriate value of $p$ should be 1.

The previously mentioned angular component is defined in terms of the real spherical harmonics:

$$y_l^m(\theta,\phi) = (-1)^m \sqrt{\frac{2l+1}{4\pi}\frac{(l-|m|)!}{(l+|m|)!}} P_l^{|m|}(\cos\theta) \begin{cases} 1 & \text{if } m = 0 \\ \sqrt{2}\cos(m\phi) & \text{if } m > 0 \\ \sqrt{2}\sin(|m|\phi) & \text{if } m < 0 \end{cases} \tag{2}$$

By combining the previous two components, we obtain the final definition of the Zernike3D basis:

$$Z_{l,n,m}(r) = \bar{R}_{l,n}^1(r)y_l^m(\theta,\phi) \tag{3}$$

### Estimating deformation fields from particles

As we explained in the previous section, the Zernike3D basis was initially formulated to be applied to 3D spaces. Therefore, it is quite direct to estimate conformational transitions from maps or atomic structures, as they live in a three-dimensional space. However, this is not the case for Cryo-EM particles, as we are intrinsically losing information along the projection direction during the acquisition process in the reduction from the three-dimensional space where the Coulomb potential of the specimen is defined to the two-dimensional space of the projection images being acquired in the microscope. In other words, conformational changes along the projection direction cannot be extracted from an individual image, since an infinite number of them would be compatible with the image information.

### Table 1 | Execution times for the Zernike3D algorithm

| Performance metrics for the Zernike3D algorithm | | | | |
|---|---|---|---|---|
| Image size | N | L | Time per-particle (s) | Time for $10^6$ particles (min–150 threads) |
| 128 | 3 | 2 | 0.1076 | 39.93 |
| 300 | 3 | 2 | 0.182 | 300.00 |

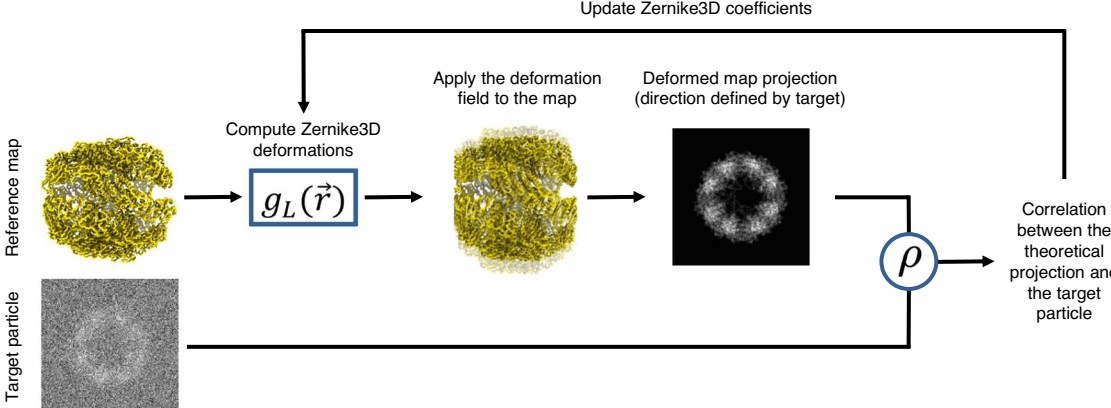

**Fig. 8 | Zernike3D workflow at particle level.** Estimation of the Zernike3D deformation fields from a particle image. The process requires the deformation of a reference map to be consistent with the dimension of the deformation field, which is defined by the set of Zernike3D coefficients. Each coefficient component is estimated by Powell's conjugate direction method, so Pearson's correlation coefficient $\rho$ between the experimental and theoretical projection obtained from the deformed map is maximized.

The algorithm we present in this work starts by computing a reference map/model (in practice, and continuing the presentation for the case of maps, it is common to either use an average map or one of the discrete classes). This map will be the origin (reference) to obtain the deformation fields from the parameters of the Zernike3D basis. The approach is summarized in Fig. 8, and it is a common procedure in optimization. In brief, it is an iterative procedure in which deformation fields are applied to the reference map and the resulting projection images are compared with the experimental ones until convergence.

Following the aforementioned method, finding the deformation field to describe the state represented by a given particle can be expressed as:

$$\max_{g_L} \rho\left(I, CP_\theta(V(r + g_L(r)))\right) \qquad (4)$$

$\rho$ being the Pearson's correlation coefficient, $I$ an experimental Cryo-EM particle, $C$ the CTF estimated for that particle, $P_\theta$ the projection operator along the 3D direction and in-plane shift defined by the parameters $\theta$, $V$ the reference volume needed to apply the Zernike3D deformation field, and $g_L$ the displacement suffered by each voxel due to the deformation field. The vector $g_L$ depends on each Zernike3D component, and it is expressed as:

$$g_L(r) = \sum_{l=0}^{L} \sum_{n=0}^{N} \sum_{m=-l}^{l} \begin{pmatrix} \alpha_{l,n,m}^x \\ \alpha_{l,n,m}^y \\ \alpha_{l,n,m}^z \end{pmatrix} Z_{l,n,m}(r) \qquad (5)$$

where the $\alpha_{l,n,m}$'s are the Zernike3D coefficients. The previous coefficients determine the contribution of each component of the basis to the deformation field.

The parameters $N$ and $L$ determine the maximum degrees of the Zernike polynomials and spherical harmonics. Therefore, they will determine the accuracy of the deformation fields: higher values will result in sharper and more accurate deformation fields, at the expense of increased execution times. By default, the two previous parameters are set to $N = 3$ and $L = 2$, which should be enough to avoid overfitting and get meaningful and accurate deformation fields in most cases. Nevertheless, the parameters can be manually set by the user in case higher accuracy is desired.

The maximization of Eq. 4 is achieved through a Powell's conjugate direction method starting from an initial guess of $\alpha_{l,n,m} = 0$ for all indices $l,n,m$ and directions $x,y,z$ (that is, no deformation). Thanks to the optimization method and the procedure described in Fig. 8, it is

possible to find the different component contributions $\alpha_{l,n,m}$ such that the deformation field to be applied to the reference map $V$ leads to a conformational state compatible with the particle $I$.

In order to avoid possible overfitting during the Powell search of the Zernike3D coefficients, an extra regularization term is added to Eq. 4:

$$\max_{g_L} \rho\left(I, CP_\theta(V(r + g_L(r)))\right) + \lambda_1 \int |g_L(r)|^2 dr \qquad (6)$$

The additional regularization term accounts for the total deformation the reference map has suffered after applying the estimated deformation field. Depending on the value of $\lambda_1$, the optimization search will be allowed to explore minima leading to a larger or smaller deformation, so it is recommended to set it at a low value to avoid overfitting without compromising the minima search. The user can choose the value of $\lambda_1$ to be applied to a specific dataset. We recommend selecting a value belonging to the range [0.01, 0.001] to avoid undesired results during the optimization process.

It is worth mentioning that the Zernike3D algorithm does not require a minimum number of particles to be executed, as the deformation fields are estimated for every particle. Therefore, it is possible to process datasets coming from a consensus or other cleaning methods[19], whose parameters are more accurately estimated but have fewer particles overall.

### Deformation field consistency along the projection direction

In this work, we estimate 3D deformation fields from 2D images. However, the information limitations introduced above make this procedure conceptually challenging. Indeed, if we compare the information stored in a projection and a map, it would be possible to check that we have identical information as long as we restrain the comparison to the projection plane where the image exists. In spite of that, the image has an intrinsic loss of information in the projection direction, as we are collapsing the map information stored along this direction.

Following the previous reasoning, the deformation field is well-defined across the projection plane, but it is ill-defined along the projection direction. This implies that there are infinite ways of deforming a map along the projection direction defined by particles so that the projection of the deformed volume is still consistent with the particle. Therefore, the Powell optimization proposed previously might find different solutions for each particle along the projection direction. Moreover, this inconsistency might lead to the global optimization process being more prone to get trapped in local minima,

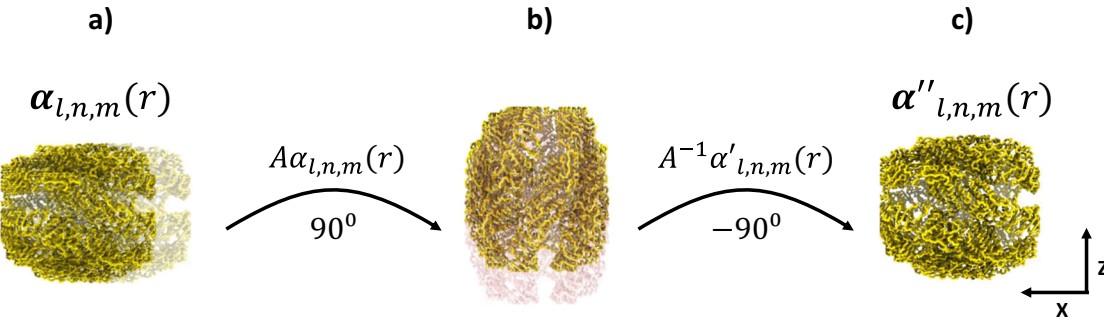

**Fig. 9 | Projection direction correction workflow.** Procedure to cancel excessive displacements associated with the projection direction in a deformation field defined by the coefficients $\alpha_{l,n,m}(r)$. **a** Representation of a deformation field with over-deformation along the X direction (projection direction). **b** The over-deformed volume and the associated deformation coefficients are rotated so that the projection direction matches the Z axis. At this point, it is possible to completely cancel the Z component on the rotated coefficients, as they only contribute to the over-deformation of the deformed map. This leads to the set of coefficients $\alpha''_{l,n,m}(r)$. **c** The modified coefficients are rotated back to the original position defined by the map's grid. In this way, a new set of coefficients $\alpha''_{l,n,m}(r)$ are obtained, defining a deformation field consistent along the projection direction (X axis).

resulting in wrong estimations of the conformational landscape. An example of the undesired effect generated by not considering the missing information along the projection direction is provided in Supplementary Fig. 3.

Ideally, the best solution to the previous problem would be to drive the optima search so that all particles are not deformed along their respective projection direction. For this reason, a sensible choice would be to completely restrict the deformation along the projection direction.

One possible way to achieve this is to include different regularization terms restricting excessive deformations. Nevertheless, it would be challenging to find the weights needed for each regularization term, especially along the projection direction, a situation that introduces a new parameter quite difficult to estimate in the process.

The Zernike3D method overcomes that obstacle by taking advantage of the properties of the basis to altogether remove any deformation along the projection direction defined by a particle, either during the optimization or after it. As we showed in ref. [10], the Zernike3D basis is closed under rotations. Thus, it is possible to rotate the Zernike3D coefficients towards a different reference frame as follows:

$$Ag_L(A^{-1}r) = \sum_{n=0}^{N}\sum_{l=0}^{L}\sum_{m=-l}^{l} A\begin{pmatrix} \alpha^x_{l,n,m} \\ \alpha^y_{l,n,m} \\ \alpha^z_{l,n,m} \end{pmatrix} \bar{Z}_{l,n,m}(A^{-1}r) \qquad (7)$$

$A$ being a rotation matrix. Therefore, it is possible to rotate the Zernike3D coefficients according to the angular information of the particle. For example, we can rotate the coefficients so that the Z direction of the new frame is effectively aligned with the projection direction of a particle. Then, we can cancel the rotated coefficients associated with the Z-axis in this new frame, as those only contribute to the deformation field along the projection direction.

However, it is essential to note that the previous property only holds in a continuous space. Hence, the basis is not closed under rotations due to the discretization and sampling of the space into voxels. Thus, the rotated coefficients cannot be applied to the volume, as the reference frames are entirely different. Instead, we can unrotate the modified Zernike3D coefficients, so their reference frame matches again with the reference map. Thus, it is possible to fully remove the deformation along the projection direction by combing all the previous steps, making all the deformation fields consistent, and avoiding solutions with unrealistic deformations. The whole procedure is summarized in Fig. 9.

Note that we also use a global regularization term in our optimization approach, as previously indicated, but it is global and does not differentiate among projection directions. In our experience, the previously stated "consistency principle" introduced by mathematically clear handling of the lack of information along individual projection directions is quite important factor in our quest for estimating deformation fields, and it represents a clear advantage of the Zernike3D approach.

## Zernike3D-based ART reconstruction algorithm

In general, 3D reconstruction algorithms start from the principle that we have a set of projections coming from a homogeneous set of particles. However, this assumption no longer holds for those macromolecules exhibiting large degrees of freedom. Therefore, it is not a surprise that molecular motions are a well-known source of blurring artifacts arising when reconstructing a Cryo-EM map from a set of Cryo-EM images. As a consequence, correcting the motions present in a particle image will be expected to boost the resolution and resolvability of blurry areas in Cryo-EM maps.

The per-particle estimation of the deformation fields by the Zernike3D basis can be effectively applied to correct molecular motions, aiding the reconstruction process with flexible information. To that end, we developed an ART-based (Algebraic Reconstruction Technique) reconstruction algorithm that uses the Zernike3D deformation fields to improve the final quality of motion-related blurry areas.

A detailed description of ART and its application in Cryo-EM can be found at ref. [20]. Here it suffices to say that ART finds the map whose projections are compatible with the experimental data through an iterative process of the form:

$$V^{(k+1)}(\boldsymbol{r}) = V^{(k)}(\boldsymbol{r}) + \lambda P_H^*(P_H V(\boldsymbol{r}) - I_k(\boldsymbol{s})) \qquad (8)$$

$V$ being the reconstructed volume, $\lambda$ the ART relaxation factor, $P_H$ the projection operator, and $P_H^*$ its adjoint operator, $I_k$ the experimental image used at the $k$-th iteration, $r$ a 3D coordinate, and $s$ a 2D coordinate. The previous equation refers to the update to be applied to the reconstruction associated with a single image, although the algorithm will iterate over the whole particle dataset applying the previous correction to achieve the final reconstruction.

One advantage of ART over other reconstruction methods is that it can be easily modified to include new information to be taken into account during the iterative reconstruction process. Thus, it is possible to modify the previous equation by adding the deformation field previously estimated:

$$V(r)^{k+1} = V(r)^k + \lambda P_H^*(P_H(V(r + g_L(r)) - I_k(s)) \qquad (9)$$

$g_L(r)$ being the displacement at a given 3D position computed through Eq. 5.

By introducing the displacements $g_L$ into the ART algorithm, we are improving the correction value that will be applied to $V$ at each iteration, as the difference between the theoretical and experimental images is taken based on the conformational change present in the particle. Thus, the reconstruction process can generate more meaningful solutions for areas subjected to significant motions.

## Moving Zernike3D coefficients through different resolutions

One of the main issues arising when working with Cryo-EM particles is the low signal-to-noise ratios that they exhibit. Although the average of a very high number of images overcomes that problem, the procedure also mixes several conformational changes at the same time. Therefore, the estimation of continuous flexibility is usually done directly on particle images, even if conditions are far from ideal.

To estimate motions more efficiently, it is common to filter the particles at a given resolution to increase the signal-to-noise ratio. Similarly, it is possible to downsample the images after the filtering process to improve the performance of the estimations.

However, the Zernike3D coefficients $\alpha_{l,m,n}$ have a strong dependency on the size of the volume under study. This implies that the coefficients computed from a downsampled map cannot be directly applied to the original volume and vice versa.

Luckily, it is possible to move a set of Zernike3D coefficients to a different resolution similar to the procedure described in previous sections. By downsampling a map, we scale its space by a given factor $k$. Therefore, the relation between two vectors with the same direction in the previous two spaces is:

$$r_o = k r_d \tag{10}$$

$r_o$ and $r_d$ being the vectors associated with the original and downsampled spaces, respectively.

We can express the earlier two vectors based on the components of the Zernike3D basis as follows:

$$\sum_{n=0}^{N} \sum_{l=0}^{L} \sum_{m=-l}^{l} \alpha_{l,n,m}^{o} \tilde{Z}_{l,n,m}(\boldsymbol{r_o}) = k \sum_{n=0}^{N} \sum_{l=0}^{L} \sum_{m=-l}^{l} \alpha_{l,n,m}^{d} \tilde{Z}_{l,n,m}(r_d)$$
$$= \sum_{n=0}^{N} \sum_{l=0}^{L} \sum_{m=-l}^{l} k\alpha_{l,n,m}^{d} \tilde{Z}_{l,n,m}(k^{-1}\boldsymbol{r_o}) \tag{11}$$

Thanks to Eq. 11, it is possible to show that the scaling relation existing between the vectors $x_o$ and $x_d$ is shared by the corresponding Zernike3D coefficients:

$$\alpha_{l,n,m}^{o} = k\alpha_{l,n,m}^{d} \tag{12}$$

This leads to a very convenient and straightforward conversion to use coefficients estimated on low-resolution images in the original high-resolution maps.

## Merging embeddings of different nature

Our previous work[10] showed that the Zernike3D basis could effectively study the continuous heterogeneity of Cryo-EM maps and atomic structures converted to electron densities.

Similarly, we have proven in the previous sections the applicability of this same tool to a set of Cryo-EM particles. In all cases, the estimation of the deformation fields represented by the Zernike3D coefficients is comparable, meaning that we are translating the information of the three main Cryo-EM entities (maps, atomic structures, and particles) to a common framework or space defined by the coefficients $\alpha_{l,m,n}$.

Translating maps, structural models, and particles to a common framework opens interesting possibilities and advantages, such as studying and comparing discrete and continuous heterogeneity or addressing how well simulated and experimental data correlate.

## Reporting summary

Further information on research design is available in the Nature Portfolio Reporting Summary linked to this article.

## Data availability

The datasets analyzed with the Zernike3D algorithm and ZART are publicly available in EMPIAR under the entries: 10028 [https://doi.org/10.6019/EMPIAR-10028], 10514 [https://doi.org/10.6019/EMPIAR-10514], 10516 [https://doi.org/10.6019/EMPIAR-10516], and 10180 [https://doi.org/10.6019/EMPIAR-10180]. The phantom dataset processed in the Supplementary Material is available in GitHub in the repository Zernike3D_Phantom_Data [https://zenodo.org/badge/latestdoi/541505536][21].

## Code availability

The Zernike3D algorithm has been implemented in Xmipp[22] and it is available through Scipion[12] under the plugins "scipion-em-xmipp" and "scipion-em-flexutils". The protocols corresponding to the algortihms described in this manuscript are "flexutils - angular align - Zernike3D" and "flexutils - reconstruct ZART".

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

## Acknowledgements

Funding is acknowledged from the Ministry of Science and Innovation through grants: Grant PID2019-104757RB-I00 funded by MCIN/AEI/10.13039/501100011033/ and "ERDF A way of making Europe", by the "European Union; the 'Comunidad Autonoma de Madrid' through grant S2017/BMD-3817; and the European Union (EU) and Horizon 2020 through grants EnLaCES (H2020-MSCA-IF-2020, Proposal: 101024130 to J.M.K.), HighResCells (ERC-2018-SyG, Proposal: 810057) and iNEXT-Discovery (Proposal: 871037). This work has also been supported by the NIH/NIGMS (No. 1R01GM136780-01 to RRL) and AFSOR FA9550-21-1-0317.

## Author contributions

D.H. developed and tested the Zernike3D and ZART algorithms presented in the manuscript. R.R.L. helped with the understanding and demonstration of the mathematical properties of the Zernike3D basis. J.M.K. and M.M. helped with the understating of the results and the preparation of the data. A.J.M. helped with the implementation of the Zernike3D algorithm in Scipion. D.M., D.S., and J.F. helped with the optimization of the codes. C.O.S.S. and J.M.C. jointly supervised this work.

## Competing interests

The authors declare no competing interests.
