## [Peer Review File · Nature Communications]

Estimating conformational landscapes from Cryo-EM particles by 3D Zernike polynomialsREVIEWER COMMENTS

Reviewer #1 (Remarks to the Author):

Here Herros et al. present an innovative new approach for analysing structural heterogeneity using Zernicke Polynomials as basis for modelling conformational heterogeneity. The method is genuinely innovative compared to other current approaches and may be very valuable for the field. However, I have a few concerns that should be addressed before publishing.

1) I communicated with the editor, requesting the software used here to get my impression from the software. I was told that this would not be straightforward to install and use at the moment, which is why it was not made available. However, I believe the software is essential, and the manuscript can only be published with available software. This should be in the best interest of the authors as only with useable software will this manuscript get the deserved attention. I thus urge the authors to make this their highest priority.

2) There is quite a zoo of tools analysing heterogeneity available (e.g. cryodrgn, 3dva, e2gmm, alphacryo. For one, it would be good if the differences in approach and computational performance could be discussed, but it would also be great if the results could be compared. The spliceosomal B complex (EMPIAR 10180) has become a standard for this, and each approach uses this to show consistency. As there is no ground truth, this is, in my view, the best comparison that could be taken.

3) Compositional heterogeneity is a significant part of heterogeneity in cryo-EM datasets. I am not sure how well the model is able to account for these. Again for analysing, there seems to be a standard dataset, an assembling ribosome (EMPIAR - 10076). Again it would be great to analyse this dataset to compare it with other approaches.

4) Most of the figures are unclear or don't contribute much to the understanding of the manuscript. For instance, I have difficulty understanding what figure 3 should tell the reader. Also, the text doesn't provide clues that would support this. Figure 4 also is not easy to read. For instance, I don't know what the black line should tell me. As it belongs according to the text and caption in Figure 3 the figures should be combined to make it clearer. Also, Umeps itself have little information value without additional descriptions of clustered points or at least more than two points. Figure 6 could also be improved significantly. For instance, the 3DVA paper from the cryosparc authors makes a good effort by using much more contrasting colours. I would heavily recommend improving all figures in presentation, clarity and content. Unfortunately, I have to say all the other tools for analysing heterogeneity did a much better job in this regard.

5) Also the math can be much clearer. Various parameters/variables are not explained and also some steps are not really followable. Even when standard definitions like the Zernicke polynomials are used all parameters should be explained as the majority of the readers of this manuscript probably don't remember them by heart. The authors should make some effort to give the interested reader the chance to follow with general scientific background.

6) Finally, it would be interesting to see how interpretable the final maps are. How is the approach performing at higher resolution? Can the result maps easily fitted by pdb files and do the results make sense?

I think this can be a very valuable approach; however, the presentation needs to be significantly improved, and some tests that have become a standard in the field should be performed before I wholeheartedly recommend publication.

Reviewer #2 (Remarks to the Author):

See attached PDF.

Estimating conformational landscapes from Cryo-EM particles by 3D Zernike polynomials – Reviewer’s report

June 20, 2022

1 Key results

The paper proposes an algorithm for analyzing heterogeneity in cryo-electron microscopy data. The problem considered in the paper is of utmost importance, with good algorithms having far-reaching consequences on our biological understanding.

The paper offers two contributions. First, it proposes an extension of an existing method (the authors’ previously published Zernike3D method) to per-image analysis. In short, they propose a method to model the deformation in each input image relative to a reference model. Second, the authors propose a new reconstruction algorithm (ZART) which compensates for the per-image deformation during reconstruction.

2 Validity

The first contribution the paper makes (extension of the Zernike3D method) is somewhat incremental. This contribution is not accompanied by exceptional theoretical or biological results.

For the second contribution (a new reconstruction algorithm), I failed to understand a key point. The second equation on page 7 seems to imply that only a single image is involved in the reconstruction of the volume. In such a case, it is unclear to me how is it possible to extract accurate three-dimensional information from a single image. If multiple images are involved in the reconstruction, then I failed to understand that from the text.

3 Significance

The proposed algorithm may turn out to extremely useful and impactful in the field of cryo-electron microscopy data analysis. However, in my view, its presentation (both the method and the results) should be improved (see details below).

4 Results

1. For the data set in Section 3.1, it is unclear if its contains also noise, shifts, and CTF. Moreover, it consists of 225 images, so it is unclear how high resolution structures can be

obtained, even without heterogeneity.

2. Section 3.1 page 9, the authors write “An interesting effect can be observed from this result: images are arranged around a 3D conformation under ideal conditions. The dispersion of the images with respect to their corresponding ground truth coefficients depends on how much error is committed due to the information missed during the projection process. Therefore, particles generated from the reference map have a low error (as we are already close to the ideal coefficient). However, as we move along the conformational transition, errors will increase due to the limitation of working with images, leading to a more significant dispersion.” I failed to see how the conclusions in this paragraph can be drawn from Figure 3.
3. In Section 3.2, it is not mentioned if the reprocessing was executed with or without heterogeneity analysis.
4. Section 3.2 discusses improvements in resolution, but without providing quantitative measure (“no numbers”).
5. There is no reference figure to show the advantage presented in Figure 2. For example, an embedding based on steerable PCA of the images. This will support the claim that the new methods indeed extracts useful information.
6. Why are two components enough in Figure 2? Maybe more components will allow for a better separation. It is impossible to say how many components are requires, since the eigenvalues of the covariance matrix are not given.
7. For Figure 4, are there any quantitative measures of similarity? In particular, given the low resolution of the middle map, I failed to validate the conclusions in the caption of the figure. Maybe a more guided explanation would help (such as marking the structural difference)?
8. There is no color bar in Figure 5 to assess the magnitude of the deformations.
9. Caption of Figure 6, I failed to understand the sentence “The deformation field corresponds to the mean coefficient associated to the selected particles”.
10. Regarding Figure 6, there are differences between the structures generated by Zernike3D and by CryoSparc. How are those explained? What is the “correct” structure (is there a ground-truth)? What are the resolutions? (in particular, what is the reference map and what is its resolution?)

5 Suggested improvements

1. The optimization model proposed by the authors (first equation on page 4), does not consider CTF, which is not mentioned elsewhere in the paper. Yet, given the presented results, I assume that authors have considered it in their implementation. It would contribute to the completeness of the presentation if they at least mention that.
2. The method is based on processing each input individually. Given, the high levels of noise, the amount of information one can extract from a single image is limited. The authors have not provided information regarding the amount of noise their algorithm can tolerate. It would be helpful to include at least one challenging dataset.

3. On page 5 the authors write “Moreover, this inconsistency might lead to the global optimization process being more prone to get trapped in local minima, resulting in wrong estimations of the conformational landscape.” I could not find any support for this claim in the paper (not analytical nor numerical).
4. Computational and numerical aspects are missing from the paper, such as convergence speed, computational complexity, number of iterations, convergence criteria, and running times.
5. It is not discussed how the parameters N and L were chosen.
6. On page 11, what does it mean “that the Zernike3D coefficients are computed exclusively based on geometrical considerations”?
7. I failed to find in the text claims supporting the last paragraph of the Conclusions.

We have followed all considerations made by the referees, including all the new experiments proposed, and the manuscript has indeed increased its quality thanks to them.

We have included substantially more mathematical details of the core part of estimating the deformation fields and the understanding of the continuous flexibility landscapes. However, when we started to do so with the ZART algorithm, we realized that the description was too detailed and lengthy, and also that the algorithm was more general than its application to Cryo-EM. In this way, we have kept the description and results of ZART in the revised manuscript as it was in the original one to be self-contained, but we have also written a much more detailed and lengthy description of all ZART capabilities in the form of a technical manuscript that will be submitted elsewhere (we are including it in this review and we will be submitting it shortly to BioRxiv, in the first place, to assure fast complementation of the two works). We hope that this way of proceeding will simplify the review process at the same time that it will provide for much more information about the algorithms that in our initial presentation in the original work.

Reviewer 1

Here Herreros et al. present an innovative new approach for analysing structural heterogeneity using Zernike Polynomials as basis for modelling conformational heterogeneity. The method is genuinely innovative compared to other current approaches and may be very valuable for the field. However, I have a few concerns that should be addressed before publishing.

Comments:

1) I communicated with the editor, requesting the software used here to get my impression from the software. I was told that this would not be straightforward to install and use at the moment, which is why it was not made available. However, I believe the software is essential, and the manuscript can only be published with available software. This should be in the best interest of the authors as only with useable software will this manuscript get the deserved attention. I thus urge the authors to make this their highest priority.

We completely agree with the reviewer about the availability of the software and we have a very well documented track record in this respect. Therefore, we have devoted most of our

efforts to make it available through Scipion and Xmipp, so it can be accessed freely and easily by any user. Consequently, the software described in the manuscript, together with some other tools that, in our opinion, will be of great value to analyze the resulting landscapes, are already available in the development channels of Scipion and Xmipp. Therefore, it should be possible to easily install all the software following the guide available here. Devel mode can be activated during the installation by adding the flag `-devel` and removing `-noXmipp` when calling the installer. In addition to Scipion and Xmipp, it will be needed to install as well the plugin scipion-em-flexutils, which contains most of the flexibility tools we are developing.

In addition, the release of the code in the stable versions of these softwares will occur during the next Scipion and Xmipp releases, which are planned for the following months.

We have extended the manuscript with a new section “Code Availability” describing the plugins and programs to be installed within the Scipion framework in order to execute all these tools. The new text included in the manuscript is the following one:

“The Zernike3D algorithm has been implemented in Xmipp (21) and it is available through Scipion 3.0 (12) under the plugins *scipion-em-xmipp* and *scipion-em-flexutils*. The protocols corresponding to the algorithms described along this manuscript are *flexutils - angular align - Zernike3D* and *flexutils - reconstruct ZART*.”

2) There is quite a zoo of tools analysing heterogeneity available (e.g. cryodrgrn, 3dva, e2gmm, alphacryo. For one, it would be good if the differences in approach and computational performance could be discussed, but it would also be great if the results could be compared. The spliceosomal B complex (EMPIAR 10180) has become a standard for this, and each approach uses this to show consistency. As there is no ground truth, this is, in my view, the best comparison that could be taken.

We agree with the reviewer that the EMPIAR 10180 has become quite standard to assess the quality of continuous heterogeneity methods, as it presents a wide range of conformational changes.

Following the reviewer advice, we have processed the dataset and analyze the flexibility of the EMPIAR 10180 spliceosome. The results we have obtained are discussed in the Results

section. Together with the text, we have included a new Figure and a Supplementary Video to better present and make clearer the discussion of the results.

As we discussed in the manuscript, the comparison of the Zernike3D and CryoDrgn results for this dataset leads to a similar representation of the landscape in the continuous heterogeneity regime. However, the main advantage of our approximation is that our representation of the conformational changes will have always the same resolution as the reference map for any point in the landscape. In addition, we can represent our results simultaneously both at the level of Cryo-EM maps and as structural models, which opens the unique possibility of easily combining in the future Cryo-EM experimentally based landscape with molecular dynamic simulations to extract real energetic profiles and trajectories. These comments have been made explicit in the new manuscript.

3) Compositional heterogeneity is a significant part of heterogeneity in cryo-EM datasets. I am not sure how well the model is able to account for these. Again for analysing, there seems to be a standard dataset, an assembling ribosome (EMPIAR - 10076). Again it would be great to analyse this dataset to compare it with other approaches.

The Zernike3D algorithm has been specifically designed to deal with continuous heterogeneity, so it will not be able to give meaningful results if compositional heterogeneity is not handled beforehand.

The main reason behind this is that we are computing deformation fields, which do not have the ability to generate or remove densities in the reference map, which would be needed to analyze compositional heterogeneity.

We have extended the description of the algorithm to make clearer the cases where it can be applied.

“In this work, we extend our recent Zernike3D algorithm (10) (specifically designed to deal with continuous heterogeneity) to precisely accomplish the latter task starting from CryoEM images with some unique properties, such as...”

4) Most of the figures are unclear or don't contribute much to the understanding of the manuscript. For instance, I have difficulty understanding what figure 3 should tell the reader. Also, the text doesn't provide clues that would support this. Figure 4 also is not easy to read. For instance, I don't know what the black line should tell me. As it belongs according to the text and caption in Figure 3 the figures should be combined to make it clearer. Also, Umaps itself have little information value without additional descriptions of clustered points or at least more than two points. Figure 6 could also be improved significantly. For instance, the 3DVA paper from the cryosparc authors makes a good effort by using much more contrasting colours. I would heavily recommend improving all figures in presentation, clarity and content. Unfortunately, I have to say all the other tools for analysing heterogeneity did a much better job in this regard.

We have modified all the Figures in the manuscript to make them more informative and clearer.

Some of the Figures have been moved to the supplementary material, where we have added extra information about the phantom experiment to better explain and support our claims.

The color scheme in Figure 2 has been improved to increase the contrast of the maps, and we have also included an example of the possibility to apply the Zernike3D deformation fields to atomic structures.

The colormap of the Zernike3D UMAP spaces has been modified, and a color bar has been added to better explain the information provided by the coloring.

Regarding the exploration of the landscapes, we have made some Videos that should help in the understanding of the different conformational changes represented by the particles. We hope that the videos will improve the description of the landscape as requested by the reviewer.

5) Also the math can be much clearer. Various parameters/variables are not explained and also some steps are not really followable. Even when standard definitions like the Zernike polynomials are used all parameters should be explained as the majority of the readers of this manuscript probably don't remember them by heart. The authors should make some effort to give the interested reader the chance to follow with general scientific background.

We have written a "Supplementary Information" file detailing in a clearer manner the math behind the Zernike3D basis to make the manuscript self-contained:

- Section A focuses on the definition and description of the basis composition, together with the expression we use to define the deformation fields in terms of the basis.
- Section B is an extension of Section "Deformation field consistency along the projection direction" in the manuscript, with a demonstration of the closure under rotation property of the basis required to handle undefined deformations along a particle projection direction.

6) Finally, it would be interesting to see how interpretable the final maps are. How is the approach performing at higher resolution? Can the result maps easily be fitted by pdb files and do the results make sense?

Zernike3D can be applied to images at any resolution. However, and in common with most methods in the field, we normally prefer to work with down sampled or filter down images, at least in the initial steps of the algorithms. Note, however, that Zernike3D is very flexible on scaling matters of images, maps or deformation fields. In this way the user can decide in the Zernike3D Scipion protocol the size of the images to be used during the analysis.

Even if the Zernike3D deformation fields are usually estimated in down sampled images, it is still possible to apply them to the original size reference map thanks to the scaling of the deformation fields and Zernike3D coefficients. The previous property also allows to apply the deformation fields effectively to atomic structures. Note that the latter is a unique property of Zernike3D that in practice it means that the user can work with maps or structural models simultaneously. Obviously, the difference between the models can only be geometrical, following the deformation field approach, but they can be excellent starting points for

specialized fitting algorithms. We have provided an example of this application in Figure 2c and Figure 4c.

Reviewer 2

The paper proposes an algorithm for analyzing heterogeneity in cryo-electron microscopy data.

The problem considered in the paper is of utmost importance, with good algorithms having far-reaching consequences on our biological understanding.

The paper offers two contributions. First, it proposes an extension of an existing method (the authors' previously published Zernike3D method) to per-image analysis. In short, they propose a method to model the deformation in each input image relative to a reference model. Second, the authors propose a new reconstruction algorithm (ZART) which compensates for the per-image deformation during reconstruction.

The first contribution the paper makes (extension of the Zernike3D method) is somewhat incremental. This contribution is not accompanied by exceptional theoretical or biological results. For the second contribution (a new reconstruction algorithm), I failed to understand a key point. The second equation on page 7 seems to imply that only a single image is involved in the reconstruction of the volume. In such a case, it is unclear to me how it is possible to extract accurate three-dimensional information from a single image. If multiple images are involved in the reconstruction, then I failed to understand that from the text.

As the reviewer suggests, the ZART equation only refers to the update to be done to the reconstruction based on a single image. However, the ZART algorithm needs to iterate over the whole particle dataset, applying the update to all the images to achieve a meaningful reconstruction. We have made clearer this process in the main manuscripts with the following text:

“The previous equation refers to the update to be applied to the reconstruction associated with a single image, although the algorithm will iterate over the whole particle dataset applying the previous correction to achieve the final reconstruction.”

The proposed algorithm may turn out to be extremely useful and impactful in the field of cryo-electron microscopy data analysis. However, in my view, its presentation (both the method and the results) should be improved (see details below).

Results

1) For the data set in Section 3.1, it is unclear if it contains also noise, shifts, and CTF. Moreover, it consists of 225 images, so it is unclear how high resolution structures can be obtained, even without heterogeneity

The phantom dataset used in this section consists of projections without noise, shifts, or CTF. The main purpose of constructing the dataset in this way is to have an ideal scenario where we can address and evaluate the different effects affecting the landscape. We think it is good practice to analyze algorithms in a sort of “first principles” way, but we have thought that this consideration may be too detailed for many readers, and we have moved it to Supplemental Material.

Regarding the number of particles, that could be considered a negative factor, it is precisely the contrary, and this test show that we do not have a limitation in the number of particles needed in the calculation of the landscape. This situation is the opposite that the one happening in other methods relying on neural networks, where it is required to have a relatively large number of particles (~100000 or more) to calculate a landscape. However, in many practical situations, once the datasets are very well pruned or the analysis is done for the particles inside a small class, we may not have datasets with these large number of images. Since our method works on a per-particle basis, we do not have a lower bound restriction in the number of particles.

Added to Supplementary material: “Subsequently, each atomic structure was converted to Coulomb potential maps using the Electron Atomic Scattering Factors (EASFs) (5), and a projection gallery of 45 particles was created for each conformation, leading to a total of 225 particles (having such a low number of particles was done on purpose to show that we can successfully analyze small datasets where a genuine continuous flexibility is present). The previous phantom particles were not further process to include noise, shifts, or the Contrast Transfer Function (CTF) to be able to determine the lower bound of the method’s errors. In our tests with experimental data the effects of noise, shifts and CTF are obviously included.”

Added to main manuscript: “It is worth mentioning that the Zernike3D algorithm does not require a minimum number of particles to be executed, as the deformation fields are

estimated for every particle. Therefore, it is possible to process datasets coming from a consensus or other cleaning methods (19), whose parameters are more accurately estimated but have fewer particles overall.”

2) Section 3.1 page 9, the authors write “An interesting effect can be observed from this result: “images are arranged around a 3D conformation under ideal conditions. The dispersion of the images with respect to their corresponding ground truth coefficients depends on how much error is committed due to the information missed during the projection process. Therefore, particles generated from the reference map have a low error (as we are already close to the ideal coefficient). However, as we move along the conformational transition, errors will increase due to the limitation of working with images, leading to a more significant dispersion.” I failed to see how the conclusions in this paragraph can be drawn from Figure 3.

The main reason to observe this effect in the landscapes computed with this algorithm is the usage of a reference map during the computations. Since we are assessing the conformation of a given particle, we can only have an accurate estimation in the image plane, but we cannot estimate the conformation along any other direction outside of this region. This is a limitation arising any time we try to compare a 2D and a 3D space.

If the conformation of a particle is close to the reference, the previous effect will be negligible. However, as the conformational change increases, the inability to estimate real 3D information from the 2D space defined by the particle will lead to less accurate estimations translated into larger dispersions.

In order to make the paragraph mentioned by the reviewer clearer, we have improved the explanation and discussion of this effect in the supplementary material. The new explanation reads:

“Figure 1b provides valuable information about the landscapes computed with the Zernike3D approach.

- Since the method relies on a reference map, the estimation error for each particle will increase as the conformation between the particle and the reference differs. The main reason behind this effect is that we are trying to estimate a per-particle conformation. Therefore, the Zernike3D approach can use the information along the image plane to define the deformation field needed to reach a new state. However, it cannot estimate this information along the projection direction, as it has been collapsed due to the projection process. As a result, we can see a higher dispersion in Figure 3b as particle move away from the reference map.
- The Zernike3D coefficient estimated for a particle will be placed surrounding the real 3D state associated with that particle. If we focus on the white dots in Figure 1b (corresponding to the real 3D maps used to generate the projected particles), we can appreciate the previous arrangement of the per-particle Zernike3D coefficients. As expected, conformations closer to the reference will be closer to the real conformation, as the error committed along the projection direction will be smaller.

It is worth to highlight that the previous effects are only significant when the signal to noise ratio of the particles is high. Thus, in a real case scenario, the large amount of noise present in every particle will dominate.”

3) In Section 3.2, it is not mentioned if the reprocessing was executed with or without heterogeneity analysis.

The reprocessing of the dataset did not include any heterogeneity analysis apart from the one performed with the tool we proposed in the manuscript. Thus, the reprocessing consisted of a full workflow in Scipion going from the micrographs to a map of good resolution. The particles associated to this map (with their metadata information – alignments, CTF...) and the map itself were afterwards used as the inputs of the Zernike3D analysis.

We have included a new paragraph in the article with a brief explanation of the Scipion workflow followed. The new paragraph reads:

“In this work we have reprocessed that dataset inside Scipion (12), leading to a total of 50,000 particles. The workflow followed included several cleaning steps to reduce as much as possible the number of unwanted particles, followed by some consensus protocols to compare the

parameters estimated by different algorithms (angular assignation, shifts, CTF...) and keep only the particles consistently estimated.

The previous particles were subjected to the Zernike3D analysis, translating them to a set of Zernike3D coefficients. As we did in the previous experiment, we use the default maximum basis degrees ($N = 3$ and $L = 2$) for the estimation of the deformation fields. In addition, the particles were downsampled to a box size of 125 voxels to increase the performance of the algorithm. Apart from the Zernike3D analysis, the particles were not subjected to other heterogeneity workflows such as classical 3D classification."

4) Section 3.2 discusses improvements in resolution, but without providing quantitative measure ("no numbers").

Figures 3 and 7 have been modified to include a comparison of the resolution histograms associated with the resolution maps computed for the CryoSparc and ZART reconstructions. The new panel is also discussed in the manuscript to give a detailed explanation of the new results. We hope that this will provide a more quantitative discussion of the resolution improvements.

We provide below the new text added to the article:

"Added to Subsection *Conformational landscape of EMPIAR 10028 dataset*: The next step we followed in the analysis of the dataset is to use the estimated deformation fields and the particles to reconstruct a higher resolution map by correcting the conformational changes of each image with ZART. The comparison between the map reconstructed with CryoSparc (15) and our new reconstruction algorithm is shown in Figure 3a. The comparison of the two maps shows a clear improvement at both, the level of maps (a) and slices (b), in the moving and still areas of the molecule. In order to make a more quantitative comparison of the maps, we computed the local resolution histograms of both reconstructions, which are compared in Figure 3b. Similarly to the visual inspection of the maps, the resolution histograms confirm the improvement in local resolution, being the average resolution of ZART pushed 1.01Å compared to the mean resolution of CryoSparc.

Added to Subsection SARS-CoV-2 spike one RBD up conformational landscape: The next step we followed in the analysis of the dataset was to use the estimated deformation fields and the particles to reconstruct a higher resolution map by “undoing” the conformational changes of each image. The motion-corrected map reconstructed with ZART is provided in Figure 7a. As expected, the information available in the deformation fields leads to a better resolvability of the moving areas of the spike (the RBDs and N-terminal Domains (NTDs) for this specific case), increasing the local resolution of these regions. Figure 7b shows a comparison of the local resolution histograms associated with the maps shown in Figure 7a. The correction of the per-particle conformational changes leads to a significant increment of the local resolution in the case of ZART, thanks to the reduction of the motion induced blurring present in the CryoSparg reconstruction.”

5) There is no reference figure to show the advantage presented in Figure 2. For example, an embedding based on steerable PCA of the images. This will support the claim that the new methods indeed extracts useful information.

It must be noted that in the steerable PCA embedding proposed by the reviewer, the projection orientation, in-plane translations, and the conformational heterogeneity would be all mixed, as this analysis is purely performed on the images without any knowledge of the projection direction. The variability due to the projection orientation and in-plane shifts is much larger than the one of conformational heterogeneity. Consequently, this latter would be largely diluted in the two former ones. For this reason, we think that a comparison to a steerable PCA embedding is not pertinent in this work.

6) Why are two components enough in Figure 2? Maybe more components will allow for a better separation. It is impossible to say how many components are required, since the eigenvalues of the covariance matrix are not given.

The referee is correct, of course, and we answer his/her question along three lines:

- 1) We provide below an image of the cumulative sum of the explained variance as the number of components is increased:

As it can be seen from the graph, the number of dimensions required to explained most of the original space is dataset dependent (a results which is not surprising). However, with three dimensions we are able to explain around 40% of the variance for 2 of the datasets described along the manuscript.

- 2) Although it may look like the landscapes presented along the manuscript are represented with 2 dimensions, they are actually represented in 3D (our error not to make this clearer). Indeed, in the analysis tools we have developed to interactively inspect the conformational landscape, the 3D representation is preferred in order to get an extra dimension that allows to get a better understanding of the different regions in the landscape.
- 3) We acknowledge that in future versions of our analysis tools it would be interesting to add the possibility to represent additional extra dimensions, so that the user could determine the combination of axis to be displayed in real time.

7) For Figure 4, are there any quantitative measures of similarity? In particular, given the low resolution of the middle map, I failed to validate the conclusions in the caption of the figure. Maybe a more guided explanation would help (such as marking the structural difference)?

The caption of Figure 4 (now moved to the supplementary material as Supplementary Figure 2) has been extended to briefly explain the purpose of the black line shown at the top of the maps. This is used for visualization purposes to highlight the opening and closing of the CCT complex, and how we can recover the right conformation after reconstruction thanks to the correction of the heterogeneity per particle.

The low resolution of the ZART reconstruction (middle map) comes from the limited number of projections used for this test (which consisted of a simulated dataset of 225 particles). However, the purpose of this dataset was to show that we could recover correctly the open conformation from closed particles if we considered the deformation fields, and not to show a very high-resolution map.

We have extended the discussion of this results to make clearer the purpose of this test in the supplementary material. The new text reads:

“Lastly, we applied the estimated coefficients to reconstruct a map with the new ART algorithm described previously. The main objective of this test is to determine whether deformation coefficients can be used to recover the reference volume from those images projected from the other conformations present in the dataset. Due to the reduced number of particles, the reconstructed map is not expected to achieve high resolution. Instead, the objective of this phantom dataset was to prove that ZART can "undo" per-particle conformational changes and achieve better reconstructions independently of the states they represent.

The comparison of the resulting reconstruction and the reference volume is shown in Figure 2. The figure shows that the new ZART reconstruction method can revert properly image deformations, recovering the different reference volume from them.”

8) There is no color bar in Figure 5 to assess the magnitude of the deformations.

Figure 5 (now Figure 1) has been modified to include a color bar showing the magnitudes of the deformation in Angstroms. Similarly, we have modified Figure 8 (now Figure 6) with the same color bar to make the colormaps in the landscapes more informative.

9) Caption of Figure 6, I failed to understand the sentence “The deformation field corresponds to the mean coefficient associated to the selected particles”.

We agree with the reviewer that the sentence was not clear enough and might mislead in the explanation of the Figure.

The points we took from the landscape are those marked in orange and white in Figure 1. These points can be now interactively selected in a new application we have developed to explore in real time the Zernike3D landscapes and that will be available in Scipion together with the Zernike3D and ZART software. In addition to the selection of point in the landscape, the new tool will also store the closest particles surrounding that point in the landscape, so the user can use them to do refinements or any other analysis in the Scipion project. Therefore, the manually selected conformation will become the “mean” of all the coefficients belonging to the previously generated cluster.

In order to simplify the caption of Figure 6 (now Figure 2), we have changed the sentence to:

“a) Comparison of the reference conformation required by the Zernike3D algorithm (red) and the rotated Pf80S state recovered from a homogeneous refinement with CryoSparc (blue). b) Comparison of the rotated Pf80S state recovered from the Zernike3D deformation fields (red) and the rotated Pf80S state recovered from a homogeneous refinement with CryoSparc (blue). The particles processed by CryoSparc are taken from the coefficient space area defined by the orange dot on Figure 1, and the deformation field is computed with the coefficients associated with this dot. The comparison between the maps displayed in a) and b) show that the Zernike3D conformation (b - red map) is consistent with the experimental conformation refined from the particles selected from that region of the coefficient space (blue). In addition, the application of the deformation field does not decrease the resolution of the reference map. c) Comparison of the atomic structure associated with the Zernike3D reference map

(red) and the structure deformed with the Zernike3D deformation fields (blue). Since the Zernike3D can work indistinguishably with maps, atomic structures, and particles, the rotated state can be appropriately reproduced at atomic level using the deformation fields estimated from the particles.”

10) Regarding Figure 6, there are differences between the structures generated by Zernike3D and by CryoSparc. How are those explained? What is the “correct” structure (is there a ground-truth)? What are the resolutions? (in particular, what is the reference map and what is its resolution?)

As the reviewer indicates, there are differences between the maps reconstructed by CryoSparc and ZART, which are coming from the consideration of the flexibility information computed for each particle with the Zernike3D approach. The differences mainly translate to a better definition of the ZART structures in flexible regions of the original map reconstructed with CryoSparc (an example of the difference is shown in Figure 4 of the new manuscript).

Figures 3 and 7 in the manuscript has been extended to include information about the local resolution of the two maps. For this case, the local resolution in the ZART reconstruction shows a significant improvement as observed from the histograms.

Suggested improvements

1) The optimization model proposed by the authors (first equation on page 4), does not consider CTF, which is not mentioned elsewhere in the paper. Yet, given the presented results, I assume that authors have considered it in their implementation. It would contribute to the completeness of the presentation if they at least mention that.

As the reviewer comments, CTF is indeed considered to find better deformation fields and a more meaningful conformational landscape. Actually, CTF is applied to the projections of the deformed reference volume that are compared to the experimental particles to assess whether the motion estimated is correct or not.

We have included a brief discussion in the methods section about the CTF consideration during the computation of the deformation fields.

“
$$\max_{g_L} \rho \left(I, CP_\theta \left(V(r + g_L(r)) \right) \right)$$

ρ being the Pearson's correlation coefficient, I an experimental Cryo-EM particle, C the CTF estimated for that particle, P_θ the projection operator along the 3D direction and in-plane shift defined by the parameters θ , V the reference volume needed to apply the Zernike3D deformation field, and g_L the displacement suffered by each voxel due to the deformation field. The vector g_L depends on each Zernike3D component, and it is expressed as...”

2) The method is based on processing each input individually. Given, the high levels of noise, the amount of information one can extract from a single image is limited. The authors have not provided information regarding the amount of noise their algorithm can tolerate. It would be helpful to include at least one challenging dataset.

We have included a new section in the supplementary material that focuses on the robustness of the method when facing different levels of noise. The test has been done with the same phantom data included in the Section C of the supplementary material.

“Since the Zernike3D approach estimates conformational changes based on the information of a single particle, it was interesting to test the capability of the method to determine meaningful

landscapes at different levels of noise. To that end, we simulated a larger dataset of 700 particles with the simulated CCT maps described before. Gaussian noise was posteriorly added to the particle images to simulate new datasets with varying Signal to Noise Ratios (SNRs). Each one of the previous datasets were subjected to the Zernike3D analysis to assess the capacity of the method to recover conformational landscapes under different noise levels. The resulting landscapes are provided in Figure 4.

As expected, higher noise levels will progressively decrease the quality of the landscapes. However, it is possible to see that for common SNRs in CryoEM (0.01-0.001), the Zernike3D approach is still able to recover appropriately a landscape with the expected shape and order of the conformations.”

Supplementary Figure 4: Assessment of the robustness of the Zernike3D algorithm to different levels of noise. Even at low SNRs, the Zernike3D landscape shows a meaningful shape and order of the conformations described by the phantom particles.

3) On page 5 the authors write “Moreover, this inconsistency might lead to the global optimization process being more prone to get trapped in local minima, resulting in wrong estimations of the conformational landscape.” I could not find any support for this claim in the paper (not analytical nor numerical).

We have included a new supplementary figure (Supplementary Figure 3) to clarify the claim mentioned by the reviewer. In this new Figure, we do a direct comparison of estimation of the same conformational change with and without the correction along the projection direction to avoid inconsistencies due to missing information.

We provide below the new figure and its caption for the completeness of the response letter

Supplementary Figure 3: Example of deformation field inconsistency along the projection direction. Pannel 1) shows the deformed reference map after correcting the aberrations associated with the deformation field along the projection direction. Pannel 2) shows the deformed reference map after being modified by the original deformation field (no correction along the projection direction). Pannel 3) shows the target map the deformation field is trying to approximate. As it can be seen from the Figure, if the projection direction is not handled correctly, the estimation over missing information will lead to unwanted conformational changes with exaggerated deformation magnitudes.

4) Computational and numerical aspects are missing from the paper, such as convergence speed, computational complexity, number of iterations, convergence criteria, and running times.

We have added two new tables detailing execution times (average time per-particle and execution time for 100000 images for the default parameters of the program) for the Zernike3D algorithm.

“This section is organized starting with general presentations of the Zernike3D basis and its use for the case of particles exhibiting continuous flexibility (first two subsections), and then dedicating several subsections to useful properties of our proposed method.

We also provide some metrics regarding the performance of the Zernike3D algorithm in Table 1”

Regarding the convergence, the Zernike3D approach is based on a Powell optimization algorithm included in Xmipp software, which controls the convergence internally based on its own logic.

For ZART, there is not a convergence criterion, but the user can determine the number of iterations to execute. It is also possible to decide whether all the iteration outputs are kept or not, so users can decide after the execution which iteration is better for their needs. We have seen that having more than 10 iterations does not improve the quality of the reconstruction, so we recommend setting a value between 5 and 10. By default, 10 iterations are performed.

5) It is not discussed how the parameters N and L were chosen.

The parameters L and N are chosen by the user. The higher the values specified, the more accurately motions will be estimated at the extent of increased execution times.

By default, N is set to 3 and L is set to 2, leading to an estimation of a total of 39 Zernike3D coefficients. The degree of freedom associated with these 39 values is usually more than enough to estimate the conformational changes correctly without overfitting.

The description of the parameters L and N has been extended to discuss how they are chosen.

“The parameters N and L determine the maximum degrees of the Zernike polynomials and spherical harmonics. Therefore, they will determine the accuracy of the deformation fields: higher values will result in sharper and more accurate deformation fields, at the expense of increased execution times. By default, the two previous parameters are set to $N = 3$ and $L = 2$, which should be enough to avoid overfitting and get meaningful and accurate deformation fields in most cases. Nevertheless, the parameters can be manually set by the user in case higher accuracy is desired.”

6) On page 11, what does it mean “that the Zernike3D coefficients are computed exclusively based on geometrical considerations”?

The Zernike3D basis estimates molecular motions based on the non-rigid alignment of the reference to the signal in the target particle. Therefore, these motions do not consider any stereochemistry or preliminary information regarding the molecule under study.

The previous consideration is important when Zernike3D deformation fields are directly applied to structural models, as they might induce some atomic clashes or place atoms in unstable positions. Therefore, and as a postprocessing step, we have found very useful to impose stereochemistry constraints to the deformed structures using Phenix or any other software specialized on this task to improve the quality of the models.

“In addition, the Zernike3D coefficients extracted from the conformational space can be applied simultaneously to the reference map and to a structural model traced (or aligned) from the reference. This allows obtaining a rigid fitting of the atomic positions that match the conformation of any particle in the dataset. However, it is worth mentioning that the Zernike3D coefficients are computed exclusively based on geometrical considerations, so the approximated structural models might need to be refined to correct for stereochemistry artefacts. Indeed, it should always be considered that the estimation of the deformation fields describing a given transition only depend on the rigid alignment of the reference towards the conformation represented by a given particle. Therefore, the estimated deformation field do not consider any stereochemistry constrains, which should be posteriorly imposed to avoid atomic clashes or improve Ramachandran outliers among others.”

7) I failed to find in the text claims supporting the last paragraph of the Conclusions.

We agree that it would have been useful to provide a better explanation of the ZART results in the “Conclusions” (now “Discussion” section) and in the main text. We hope that the new changes added to Figures 3 and 7 and its discussion in main text as suggested by the reviewers will improve the support of the last claim of the “Discussion” section.

REVIEWERS' COMMENTS

Reviewer #1 (Remarks to the Author):

The authors did clear out all my concerns and improved their manuscript significantly. I thus recommend publication.

Minor issue:

The contrast of the plots with dark background is bad and they are hard to see. I would recommend white background.

Reviewer #2 (Remarks to the Author):

All concerns raised by this reviewer has been adequately addressed.

There remain two minor issues with the revised version:

1. In the results of the EMPIAR 10028 dataset, the reported resolution in EMD is 3.2Å, without considering any heterogeneity. Why the results reported in the paper (after handling heterogeneity) are much worse?
2. In the section "Zernike3D basis definition", it is said that the functions are defined on the sphere. This is wrong. They are defined in the unit ball, as correctly written in the supplementary information.

Reviewer 1

The authors did clear out all my concerns and improved their manuscript significantly. I thus recommend publication.

Comments:

1) The contrast of the plots with dark background is bad and they are hard to see. I would recommend white background.

We have modified the figures having a dark background (Figures 1, 4, and 6) to make it white as suggested by the reviewer.

Reviewer 2

All concerns raised by this reviewer has been adequately addressed.

Comments

1) In the results of the EMPIAR 10028 dataset, the reported resolution in EMDDB is 3.2A, without considering any heterogeneity. Why the results reported in the paper (after handling heterogeneity) are much worse?

Originally, the heterogeneity correction implemented in ZART was only possible to be applied to the images used to compute the Zernike3D coefficients. Therefore, as the images were downsampled to improve the performance and the signal to noise ration during the estimation, the resolution of the resulting ZART corrected maps was limited.

However, the previous limitation is already solved in the currently available version of the software, which was shortly published in the stable channel of Scipion. In addition, this version was the one used in the ZART paper we shared as supplementary material, where we were able to improve the resolution of the map to 3.1A, as well as the local resolution as shown in the Figures presented in that paper.

2) In the section "Zernike3D basis definition", it is said that the functions are defined on the sphere. This is wrong. They are defined in the unit ball, as correctly written in the supplementary information.

As commented by the reviewer, there was an error in the Methods section regarding the definition of the Zernike3D basis functions, which are defined in the unit ball. We have corrected this mistake in the manuscript.